# Tools for assessing child and adolescent stunting: Lookup tables, growth charts and a novel appropriate-technology "MEIRU" wallchart - a diagnostic accuracy study

Pannapat Amy Chanyarungrojn[1]*, Natasha Lelijveld[1,2], Amelia Crampin[3,4,5], Lawrence Nkhwazi[3], Steffen Geis[3,4,6], Moffat Nyirenda[1,3,7], Marko Kerac[1]*

1 Department of Population Health, London School of Hygiene and Tropical Medicine, London, United Kingdom, 2 Emergency Nutrition Network, Oxford, United Kingdom, 3 MEIRU (Malawi Epidemiology & Intervention Research Unit), Lilongwe, Malawi, 4 Department of Infectious Disease, London School of Hygiene and Tropical Medicine, Bloomsbury, United Kingdom, 5 Glasgow University, Scotland, United Kingdom, 6 Infection Prevention and Control Unit, University Hospital Duesseldorf, Duesseldorf, Germany, 7 MRC Uganda, Entebbe, Uganda

* pannapat.chanyarungrojn1@alumni.lshtm.ac.uk (PAC); marko.kerac@lshtm.ac.uk (MK)

**Data Availability Statement:** All data can be found in the manuscript and supporting information files.

## Abstract

Stunting affects 149 million children worldwide and is a form of chronic malnutrition defined by low height-for-age. Surveys and intervention programmes depend on effective assessment and identification of affected individuals. Gold standard assessment is based on height-for-age Z-score (HAZ): HAZ <-2 defines stunting; HAZ <-3 defines severe stunting. However, a major problem for field-based programmes is that Z-scores can be time-intensive and challenging to calculate. We thus developed a novel wallchart that we have coined 'MEIRU wallchart' to easily and accurately identify stunted children and adolescents. Our study aim was to evaluate its performance and acceptability against other methods used in current clinical/field practice. We undertook a non-interventional diagnostic accuracy study in Malawi. We recruited 244 participants aged 8–19 years and determined each individual's stunting status using, in varying order: the MEIRU wallchart, traditional lookup tables, and traditional growth charts. All were compared against 'gold standard' HAZ, calculated using AnthroPlus WHO software. Local community healthcare workers performed all the assessments. The wallchart method was strongly preferred by both participants and staff. It had an overall accuracy of 95.5%(kappa = 0.91) and was faster than lookup tables by an average of 62.5%(41.4sec; p<0.001) per measurement. Lookup tables and growth charts had overall agreements of 59.4%(kappa = 0.36) and 61.9%(kappa = 0.31) respectively. At the HAZ-2 cut-off, the wallchart had a sensitivity of 97.6%(95%CI: 91.5–99.7) and specificity of 96.3% (95%CI: 92.1–98.6). We conclude that the MEIRU wallchart performs well and is acceptable for screening and identification of stunted children/adolescents by community-level health workers. It fulfils key criteria that justify a role in future screening programmes: easy to perform and interpret; acceptable; accurate; sensitive and specific. Potential future uses include: conducting rapid stunting prevalence surveys; identifying affected individuals for

**Funding:** This work was supported by the Wellcome Trust as a sub-study nested in a larger project grant for 'Stunting2Win' Study (200669/Z/16/Z to MK). This project was also supported by the London School of Hygiene and Tropical Medicine (MSc Summer Project Trust Fund to PC). The funders had no role in study design, data collection and analysis, decision to publish, or preparation of the manuscript.

**Competing interests:** The authors have declared that no competing interests exist.

**Abbreviations:** HAZ, Height-for-Age Z-score; MEIRU, Malawi Epidemiology and Intervention Research Unit; PPV, Positive predictive value; NPV, Negative predictive value; WHO, World Health Organization; ISRCTN, International Standard Randomised Controlled Trial Number; HSA, Health Surveillance Assistants; WAZ, Weight-for-Age Z-score; WHZ, Weight-for-Height Z-score; ODK, Open Data Kit.

interventions. Current field methods, lookup tables and growth charts performed poorly and should be used with caution.

## Introduction

Stunting, or linear growth failure, is the commonest form of child malnutrition worldwide, with some 149 million children under five years of age affected [1]. It is defined as a height two standard deviations below the median of an age- and sex-matched reference population, otherwise expressed as a "height-for-age Z-score" (HAZ) of <-2. Widely used and referenced in global health literature, policy and programmes, it is commonly seen as the "best overall indicator of children's well-being and an accurate reflection of social inequalities" [2]. Reducing stunting is therefore a key priority for the 2030 "Zero Hunger" Sustainable Development Goal [3].

Whilst there are many ongoing global efforts to develop interventions to prevent stunting and to treat and support those affected [4, 5], an often neglected yet critical part of any such programmes is to effectively identify affected individuals in the first place. Most focus to date has been on young children since the '1st 1000' days is a particularly sensitive period of growth and development [6]. However, there is also growing focus on adolescent nutrition and stunting [7, 8]. Adolescence is a time of rapid growth and development and it could represent an important 'second window' of opportunity for stunted children to catch-up and recover lost growth and development potential from earlier adversity [9]. Given that future interventions might target high-risk, already affected individuals, as per standard Wilson-Jungner screening test criteria, "there should be a suitable test or examination" and "the test should be acceptable to the population" [10, 11]. In current practice, stunting is evaluated by:

i. Healthcare workers measuring a child's/adolescents' height and determining his/her age;

ii. Using either a lookup table or a growth chart to determine the corresponding HAZ to classify an individual as: non-stunted (HAZ ≥-2); stunted (HAZ <-2); severely stunted (HAZ <-3); [12]

Although simple in theory, the quality of stunting assessment done in this way may vary greatly. It takes time, training, and supervision, all of which are often in short supply in resource-poor settings where stunting is common. Many factors have the potential to influence final classification including: the accuracy and precision of anthropometric equipment (height measure); measurement setting; the child's cooperation; the measurer's experience and technique; the measurer's ability to correctly 'translate' a raw height and age value into a HAZ classification [13]. Clinical assessment alone is not possible and would miss many affected individuals, not least because stunting is so common in some communities that short children are considered 'normal' [14]. A net effect of all these constraints is that field-based HAZ assessment is often a low priority and is rarely done.

In this paper we explore a novel, low-cost method of identifying stunted individuals. Inspired by the "Nabarro" weight-for-height chart [15], we designed a wallchart which we named the "MEIRU wallchart" after the Malawi Epidemiology and Intervention Research Unit (MEIRU), where we conducted this first validation study. The MEIRU wallchart is an 'appropriate technology' [16] tool to simply, safely, and precisely identify stunted individuals by assessing their height-for-age. We hypothesise that it is superior in these regards to traditional look-up tables and growth chart methods. Our vision is for it to be used in a wide range of clinical and community settings by staff with minimal training.

Our overall aim in this project was to quantify the validity and performance of the MEIRU wallchart for identification of stunted children and adolescents. Towards this, our objectives were to:

- Evaluate diagnostic performance of the MEIRU chart, measured through comparison of accuracy, sensitivity, specificity, positive predictive value (PPV), and negative predictive value (NPV) against 'gold standard' HAZ calculation;

- Quantify total time required for assessment;

- Compare the performance of the MEIRU chart against two main current methods of HAZ assessment: WHO lookup tables and growth charts;

- Assess the acceptability of the new method.

## Methods

### Ethics statement

Ethical approval was granted by the University of Malawi College of Medicine Research & Ethics Committee (reference P.06/16/1955) and the London School of Hygiene and Tropical Medicine MSc Research Ethics Committee (reference 10912). The individuals pictured in Figs 1–3 have provided written informed consent (as outlined in PLOS consent form) to publish their image alongside the manuscript.

We sought written informed consent from HSAs and written assent from the children and adolescents. We also obtained written informed consent from parents or guardians of the children and adolescents under 19 years of age.

### Study design and participants

This was a field-based, prospective, non-interventional diagnostic accuracy study. It was registered on ISRCTN: http://www.isrctn.com/ISRCTN16311596.

We collected data cross-sectionally in field sites overseen by Malawi Epidemiology & Intervention Research Unit (MEIRU) in Area 25, Lilongwe, Malawi, from 12th to 28th July 2016. Area 25 is one of MEIRU's established Demographic Surveillance Sites (DSS). It is a peri-urban area, with epidemiological and demographic characteristics typical of many other low- and middle-income countries. As such, it shares many issues and challenges faced by other communities in other countries.

Our target population of individuals being measured were older children and adolescents. They were recruited as volunteers, responding to calls from MEIRU fieldworkers who work in and know the local communities well. Before our visit to a particular location within the MEIRU DSS, our fieldworkers would discuss the study with local leaders and distribute study information sheets and consent forms for parents to review and sign.

Our target population of staff conducting the stunting measurements were local Health Surveillance Assistants (HSAs) who work with MEIRU. We chose HSAs because they are the cadre of staff who are likely to do the screening in surveys and future stunting programmes. They are community-based health-promotion and prevention staff who have secondary school education plus a few months of specialist training. They are often based in the communities they serve and are typical of community health staff who would conduct stunting assessment in other countries. To minimise any bias due to variation in individual HSA skills and experience, we worked with a different HSA on each day of fieldwork.

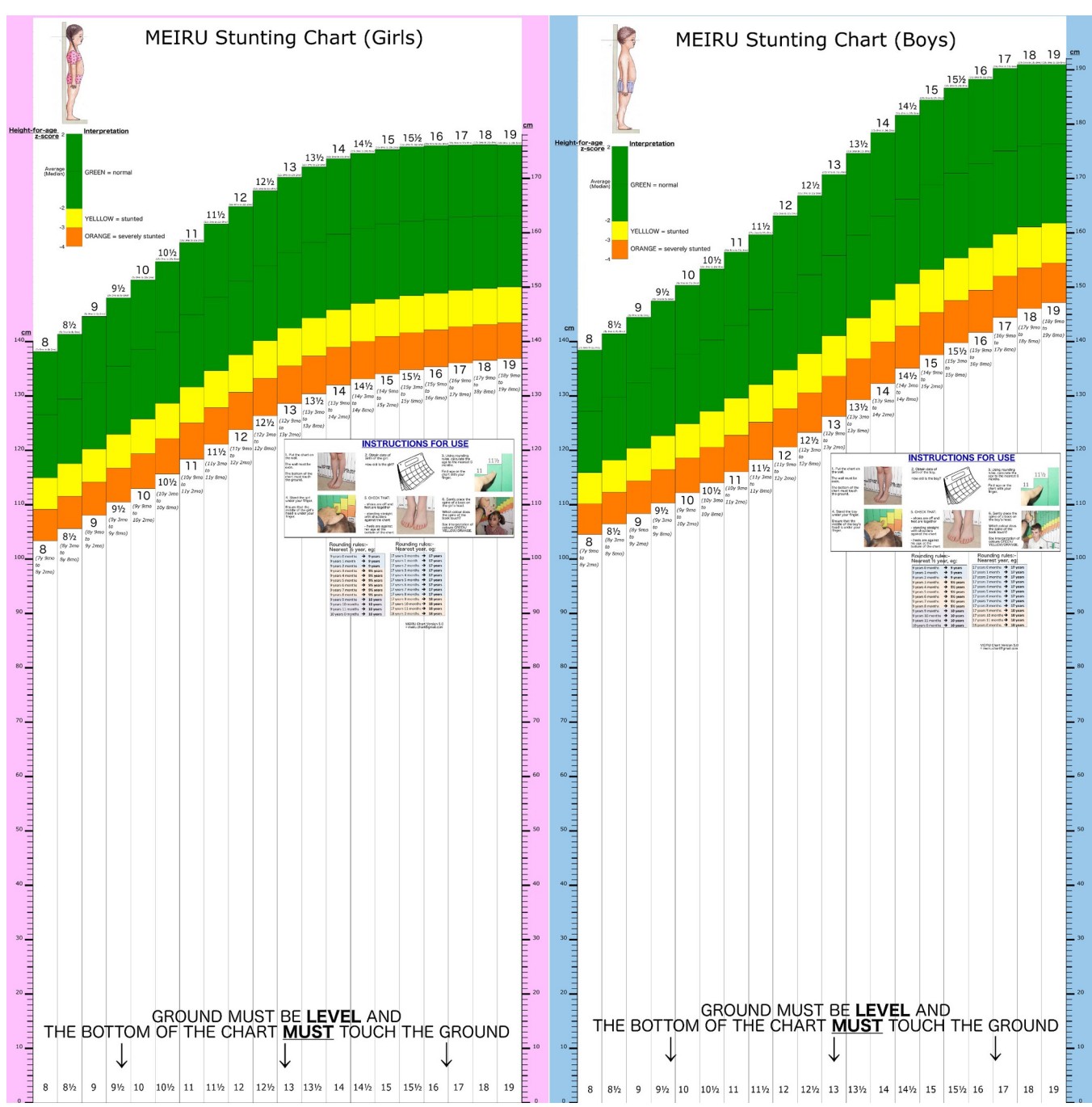

**Fig 1. The MEIRU wallcharts used for field testing.**

## Test methods

First, each adolescent's age was established by asking both age and date of birth. If the two were discordant (as per immediate calculation using our electronic data capture system), HSAs were prompted to recheck both and determine which was correct. HSAs then conducted anthropometric measurements. Finally, they translated the raw height and age values into an

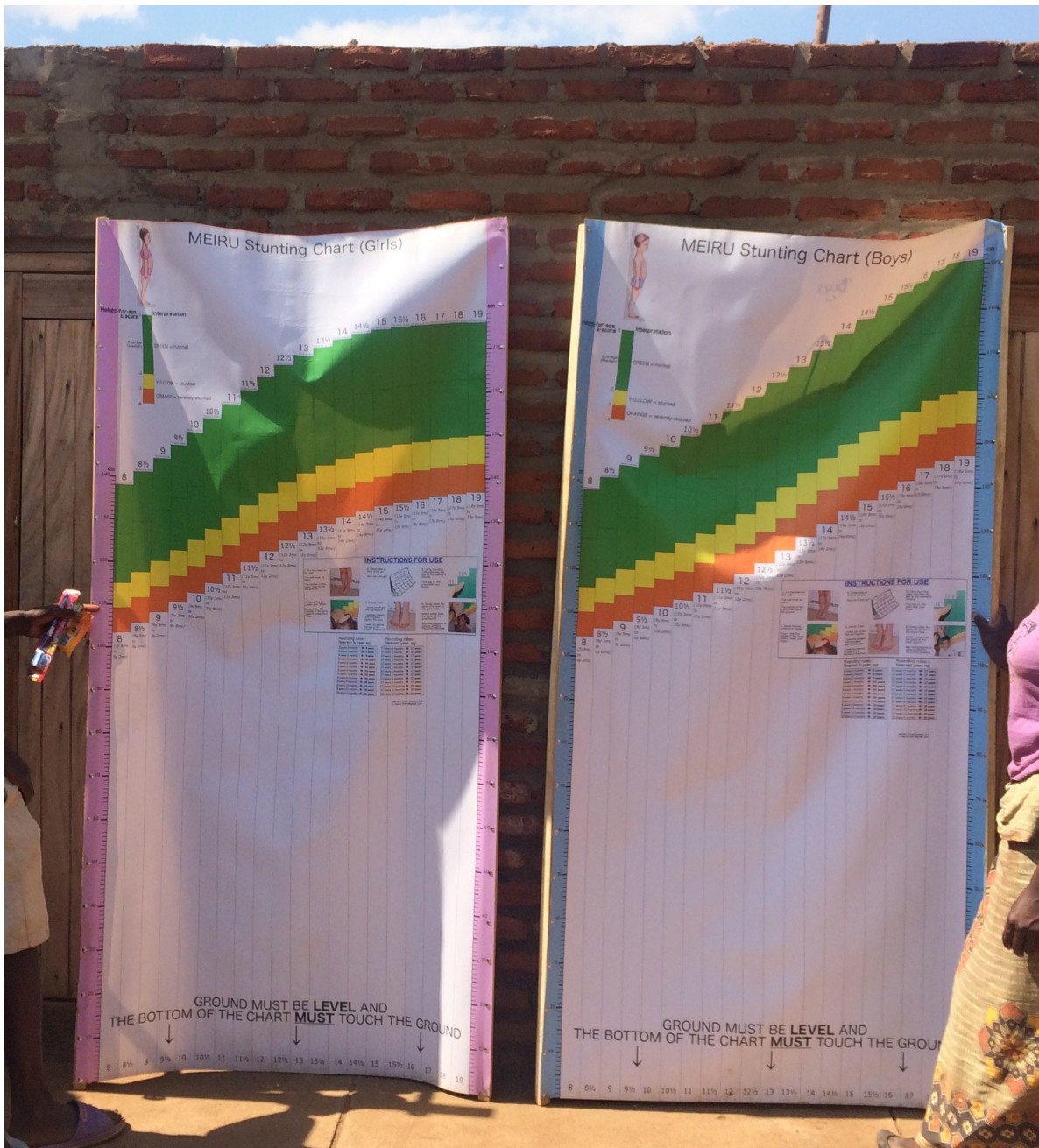

**Fig 2. Field-testing of the MEIRU wallchart showing field set-up.**

assessment of stunting status (not stunted; stunted; severely stunted). All measurements were repeated by our study team (LN).

Each individual's stunting status was assessed using three distinct methods: the MEIRU chart; World Health Organization (WHO) lookup tables; WHO growth charts. The last two assessment methods were collectively classed as "traditional" methods, as they are common current methods highlighted on the WHO website [12]. HSAs used the same raw height measurement to determine stunting status by consulting both WHO lookup tables and growth

charts. The same height measurement also went into our study database and was used to calculate exact HAZ, from which we determined the final 'gold standard' stunting status: HAZ $\geq$-2 = 'not currently stunted'; HAZ <-3 = 'severely stunted'; HAZ -3 to <-2 = 'moderately stunted'. The performance of each test method was compared against this calculated reference HAZ.

The time taken to determine stunting status using each assessment method was also measured.

Each adolescent was assessed using all methods before moving on to the next adolescent. The starting method (MEIRU wallchart or traditional methods) was alternated between odd and even participant ID numbers.

**Index method–MEIRU wallchart.**   The MEIRU wallchart is a life-size stacked bar chart (Fig 1). Each bar on the chart corresponds with a specific age (8–19 years). The sex- and age-dependent height cut-offs for each HAZ were derived from the WHO Growth Reference 2007 data [12]. Height was plotted on the vertical axis on a 1:1 scale. Each bar also had three colour-coded sections:

- Orange for severely stunted height (HAZ -4 to <-3);

- Yellow for stunted height (HAZ -3 to <-2);

- Green for normal height (HAZ $\geq$-2).

For portability and ease of use, MEIRU wallchart prototypes were printed on canvas and attached to wooden sticks held by volunteers during measurement (Fig 2).

After each adolescent's age was determined he/she stood against the corresponding age bar. Rounding rules for age were printed on the chart and helped an HSA identified the correct age-bar. Once positioned, the HSA placed the spine of a book on the adolescent's head, noting the colour on the bar where it intersected the chart (Fig 3). The colour on the bar was then matched to the child's stunting status using the interpretation key at the top of the wallchart.

**Reference standard–calculated Height-for-Age Z-score (HAZ).**   Our 'gold standard' reference in this study was computer-calculated HAZ. For this, participants' heights were measured using a Leicester stadiometer (Child Growth Foundation, UK). Following methods used in the WHO Growth Standards Multi-country Growth Reference Study [13], readings were repeated by two independent observers, with a maximum allowable difference of 0.7cm. If outside this agreement limit, both observers would re-measure. If within limits, the mean value was taken as the final height. From this, HAZs were calculated on Stata (version 14, StataCorp LP, College Station, TX), using the *WHO AnthroPlus* Stata macro package [17], an anthropometric calculator using the WHO Reference 2007 data [12].

For immediate results in the field, HSAs also identified the stunting status of each adolescent using both WHO field lookup tables and growth charts. These are commonly used for both HAZ calculation/interpretation but the same methods are also widely used for other anthropometric measures including WAZ (weight-for-age) and WHZ (weight-for-height).

To ensure fair comparison of the different methods, the standard colour schemes of both lookup tables (Fig 4) and growth charts (Fig 5) were adjusted from their WHO originals to match those of the MEIRU wallchart (which itself was based on WHO colour scheme).

**Data collection.**   Data were collected on electronic forms created using Open Data Kit (ODK, version 1.4). ODK forms only collected participant study IDs as identifiers to ensure anonymity. Start times and timestamps at critical measurement steps were also recorded by the ODK app. Whilst ODK was able to display the timestamps, the version we had was unable to store them for direct export. Therefore, timestamps were manually entered via EpiData Entry (version 2.0.9.25).

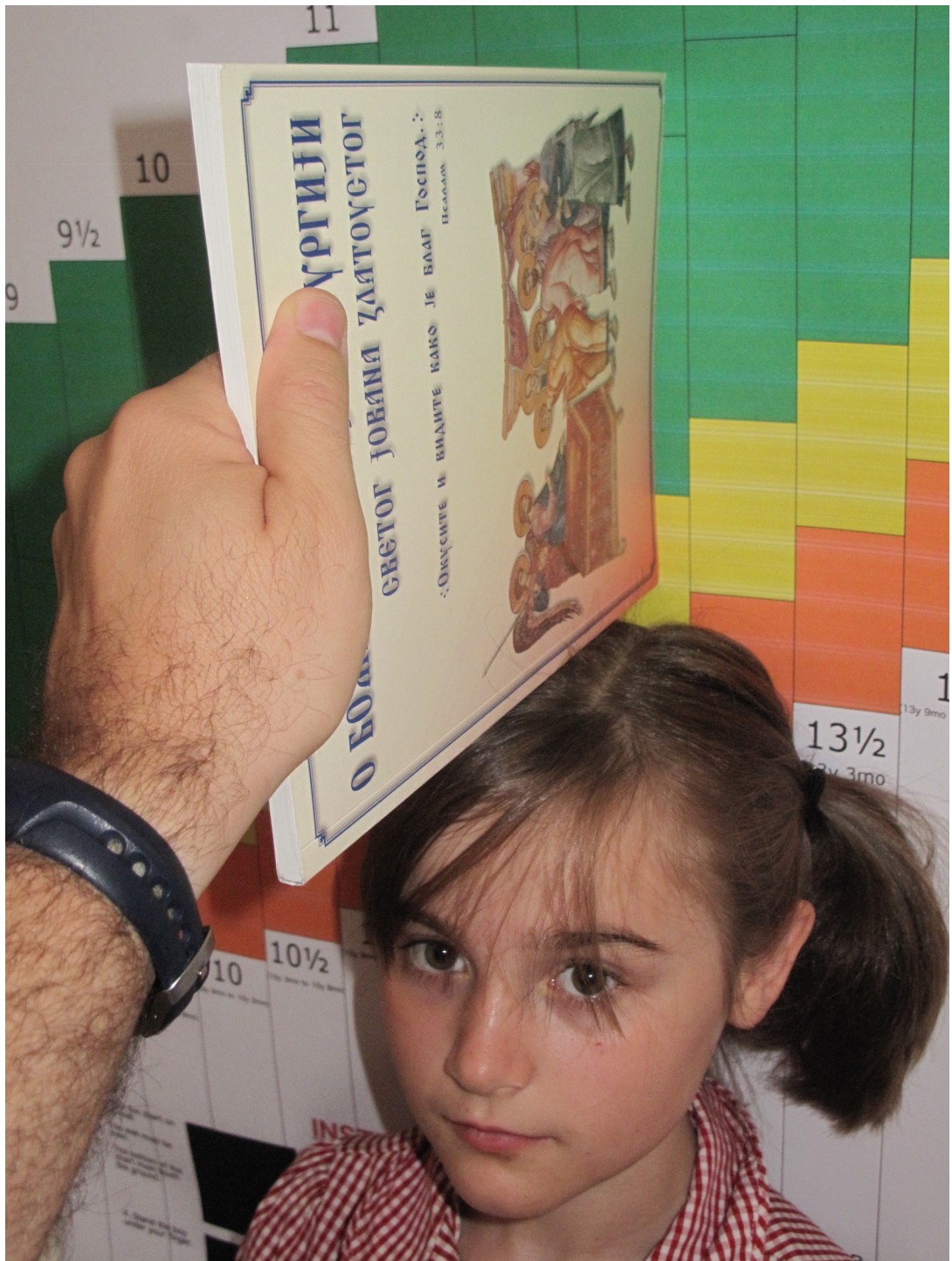

**Fig 3. Using the MEIRU wallchart with a book held upright for readings.**

## Statistical analysis

**Test methods.** We compared the following:

- Index test: MEIRU wallchart

## A

**Height-for-age BOYS**
**5 to 19 years (z-scores)**

| Year: Month | Months | -3 SD | -2 SD | -1 SD | Median | 1 SD | 2 SD | 3 SD |
|---|---|---|---|---|---|---|---|---|
| 7:7 | 91 | 108.5 | 114.0 | 119.5 | 125.0 | 130.5 | 136.0 | 141.5 |
| 7:8 | 92 | 108.9 | 114.4 | 119.9 | 125.5 | 131.0 | 136.5 | 142.0 |
| 7:9 | 93 | 109.2 | 114.8 | 120.4 | 125.9 | 131.5 | 137.0 | 142.6 |
| 7:10 | 94 | 109.6 | 115.2 | 120.8 | 126.4 | 132.0 | 137.5 | 143.1 |
| 7:11 | 95 | 110.0 | 115.6 | 121.2 | 126.8 | 132.4 | 138.1 | 143.7 |
| 8:0 | 96 | 110.3 | 116.0 | 121.6 | 127.3 | 132.9 | 138.6 | 144.2 |
| 8:1 | 97 | 110.7 | 116.4 | 122.0 | 127.7 | 133.4 | 139.1 | 144.7 |
| 8:2 | 98 | 111.0 | 116.7 | 122.5 | 128.0 | 133.9 | 139.6 | 145.3 |
| 8:3 | 99 | 111.4 | 117.1 | 122.9 | 128.6 | 134.3 | 140.1 | 145.8 |
| 8:4 | 100 | 111.7 | 117.5 | 123.3 | 129.0 | 134.8 | 140.6 | 146.4 |
| 8:5 | 101 | 112.1 | 117.9 | 123.7 | 129.5 | 135.3 | 141.1 | 146.9 |
| 8:6 | 102 | 112.4 | 118.3 | 124.1 | 129.9 | 135.8 | 141.6 | 147.4 |
| 8:7 | 103 | 112.8 | 118.7 | 124.5 | 130.4 | 136.2 | 142.1 | 148.0 |
| 8:8 | 104 | 113.1 | 119.0 | 124.9 | 130.8 | 136.7 | 142.6 | 148.5 |
| 8:9 | 105 | 113.5 | 119.4 | 125.3 | 131.3 | 137.2 | 143.1 | 149.0 |
| 8:10 | 106 | 113.8 | 119.8 | 125.7 | 131.7 | 137.6 | 143.6 | 149.5 |
| 8:11 | 107 | 114.2 | 120.2 | 126.1 | 132.1 | 138.1 | 144.1 | 150.1 |
| 9:0 | 108 | 114.5 | 120.5 | 126.6 | 132.6 | 138.6 | 144.6 | 150.6 |
| 9:1 | 109 | 114.9 | 120.9 | 127.0 | 133.0 | 139.0 | 145.1 | 151.1 |
| 9:2 | 110 | 115.2 | 121.3 | 127.4 | 133.4 | 139.5 | 145.6 | 151.7 |
| 9:3 | 111 | 115.6 | 121.7 | 127.8 | 133.9 | 140.0 | 146.1 | 152.2 |
| 9:4 | 112 | 115.9 | 122.0 | 128.2 | 134.3 | 140.4 | 146.6 | 152.7 |
| 9:5 | 113 | 116.3 | 122.4 | 128.6 | 134.7 | 140.9 | 147.1 | 153.2 |
| 9:6 | 114 | 116.6 | 122.8 | 129.0 | 135.2 | 141.4 | 147.6 | 153.8 |
| 9:7 | 115 | 116.9 | 123.2 | 129.4 | 135.6 | 141.8 | 148.1 | 154.3 |
| 9:8 | 116 | 117.3 | 123.5 | 129.8 | 136.1 | 142.3 | 148.6 | 154.8 |
| 9:9 | 117 | 117.6 | 123.9 | 130.2 | 136.5 | 142.8 | 149.1 | 155.3 |
| 9:10 | 118 | 118.0 | 124.3 | 130.6 | 136.9 | 143.2 | 149.5 | 155.9 |
| 9:11 | 119 | 118.3 | 124.7 | 131.0 | 137.3 | 143.7 | 150.0 | 156.4 |
| 10:0 | 120 | 118.7 | 125.0 | 131.4 | 137.8 | 144.2 | 150.5 | 156.9 |

## B

**Height-for-age BOYS**
**5 to 19 years (z-scores)**

| Year: Month | Months | -3 SD | -2 SD | -1 SD | Median | 1 SD | 2 SD | 3 SD |
|---|---|---|---|---|---|---|---|---|
| 7:7 | 91 | 108.5 | 114.0 | 119.5 | 125.0 | 130.5 | 136.0 | 141.5 |
| 7:8 | 92 | 108.9 | 114.4 | 119.9 | 125.5 | 131.0 | 136.5 | 142.0 |
| 7:9 | 93 | 109.2 | 114.8 | 120.4 | 125.9 | 131.5 | 137.0 | 142.6 |
| 7:10 | 94 | 109.6 | 115.2 | 120.8 | 126.4 | 132.0 | 137.5 | 143.1 |
| 7:11 | 95 | 110.0 | 115.6 | 121.2 | 126.8 | 132.4 | 138.1 | 143.7 |
| 8:0 | 96 | 110.3 | 116.0 | 121.6 | 127.3 | 132.9 | 138.6 | 144.2 |
| 8:1 | 97 | 110.7 | 116.4 | 122.0 | 127.7 | 133.4 | 139.1 | 144.7 |
| 8:2 | 98 | 111.0 | 116.7 | 122.5 | 128.2 | 133.9 | 139.6 | 145.3 |
| 8:3 | 99 | 111.4 | 117.1 | 122.9 | 128.6 | 134.3 | 140.1 | 145.8 |
| 8:4 | 100 | 111.7 | 117.5 | 123.3 | 129.0 | 134.8 | 140.6 | 146.4 |
| 8:5 | 101 | 112.1 | 117.9 | 123.7 | 129.5 | 135.3 | 141.1 | 146.9 |
| 8:6 | 102 | 112.4 | 118.3 | 124.1 | 129.9 | 135.8 | 141.6 | 147.4 |
| 8:7 | 103 | 112.8 | 118.7 | 124.5 | 130.4 | 136.2 | 142.1 | 148.0 |
| 8:8 | 104 | 113.1 | 119.0 | 124.9 | 130.8 | 136.7 | 142.6 | 148.5 |
| 8:9 | 105 | 113.5 | 119.4 | 125.3 | 131.3 | 137.2 | 143.1 | 149.0 |
| 8:10 | 106 | 113.8 | 119.8 | 125.7 | 131.7 | 137.6 | 143.6 | 149.5 |
| 8:11 | 107 | 114.2 | 120.2 | 126.1 | 132.1 | 138.1 | 144.1 | 150.1 |
| 9:0 | 108 | 114.5 | 120.5 | 126.6 | 132.6 | 138.6 | 144.6 | 150.6 |
| 9:1 | 109 | 114.9 | 120.9 | 127.0 | 133.0 | 139.0 | 145.1 | 151.1 |
| 9:2 | 110 | 115.2 | 121.3 | 127.4 | 133.4 | 139.5 | 145.6 | 151.7 |
| 9:3 | 111 | 115.6 | 121.7 | 127.8 | 133.9 | 140.0 | 146.1 | 152.2 |
| 9:4 | 112 | 115.9 | 122.0 | 128.2 | 134.3 | 140.4 | 146.6 | 152.7 |
| 9:5 | 113 | 116.3 | 122.4 | 128.6 | 134.7 | 140.9 | 147.1 | 153.2 |
| 9:6 | 114 | 116.6 | 122.8 | 129.0 | 135.2 | 141.4 | 147.6 | 153.8 |
| 9:7 | 115 | 116.9 | 123.2 | 129.4 | 135.6 | 141.8 | 148.1 | 154.3 |
| 9:8 | 116 | 117.3 | 123.5 | 129.8 | 136.1 | 142.3 | 148.6 | 154.8 |
| 9:9 | 117 | 117.6 | 123.9 | 130.2 | 136.5 | 142.8 | 149.1 | 155.3 |
| 9:10 | 118 | 118.0 | 124.3 | 130.6 | 136.9 | 143.2 | 149.5 | 155.9 |
| 9:11 | 119 | 118.3 | 124.7 | 131.0 | 137.3 | 143.7 | 150.0 | 156.4 |
| 10:0 | 120 | 118.7 | 125.0 | 131.4 | 137.8 | 144.2 | 150.5 | 156.9 |

**Fig 4.** WHO simplified field lookup tables with original (A) and amended (B) colour schemes. Adapted from: WHO 2007 [12].

- Traditional method 1: WHO lookup table

- Traditional method 2: WHO growth chart

- Reference standard: computer-calculated HAZ

All analyses were done on Stata. (version 14, StataCorp LP, College Station, TX). Participants with missing timestamps and those who were not assessed using growth charts were excluded from their respective analyses. We did exploratory descriptive analyses to visualize the distribution of the baseline characteristics. Results with each assessment method were cross-tabulated against the gold standard calculated HAZ-based status to determine the concordance between the tests. The kappa statistic and overall agreement were used as measures of agreement between each test and the reference standard [18].

We calculated standard test performance indicators: sensitivity, specificity, PPV, and NPV. Results using each assessment method were re-grouped into binary variables ("normal" and "stunted or severely stunted"), to analyse its performance around the HAZ<-2SD cut-off. We used McNemar's $\chi^2$ test as a measure of test method performance, by comparing their sensitivities and specificities separately [19, 20]. Sensitivities were compared using a 2x2 table

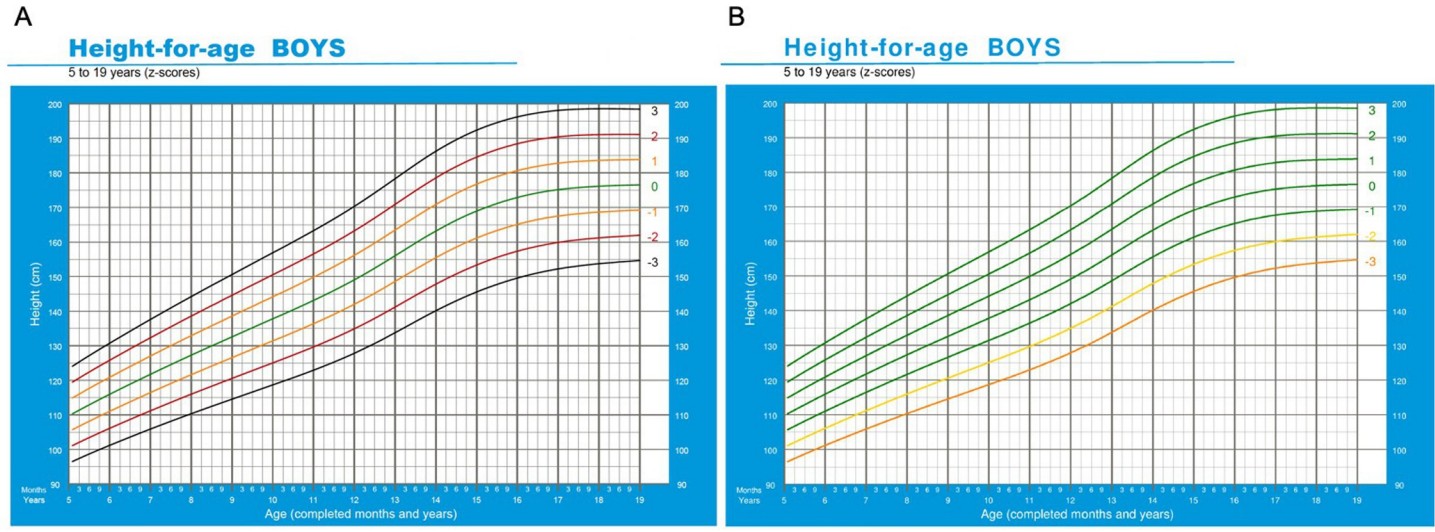

**Fig 5.** WHO growth charts with original (A) and amended (B) colour schemes. Adapted from: WHO 2007 [12].

exclusively among "stunted or severely stunted" adolescents, while specificities were compared using a 2x2 table exclusively among non-stunted adolescents [20]. The same analyses were done to calculate standard performance indicators around the HAZ<-3SD cut-off.

**Sample size.** We calculated this using McNemar's method. We aimed for 80% power and 5% type I error rate. We assumed that 5% of non-stunted adolescents will be misclassified as stunted ($p_{01}$) and 1% of stunted adolescents will be wrongly identified as non-stunted ($p_{10}$). This resulted in a target sample size of 292. This was also sufficient for a number of other related scenarios, as shown in S1 Text.

## Results

### Participants

We recruited a total of 244 adolescents (Fig 6). They were measured by 12 different HSAs, who each assessed a median of 20 individuals (IQR 17.25–25).

Index test refers to assessment using the MEIRU wallchart. Reference standard refers to calculation of exact HAZ, the current gold standard.

Their median age was 11.3 years (IQR = 9.5–13.1). 106 (43.4%) were male. Most of them, 134 (54.9%) were recruited from Chikanda, as this was a large village in our study area. S1 Table shows their demographic characteristics in detail.

Table 1 shows the distribution of adolescents who were normal, stunted and severely stunted for each method used. Based on gold standard HAZ calculation, 22.1% (54/244) of adolescents were stunted and 11.5% (28/244) were severely stunted.

Table A in S2 Text illustrates the distribution of normal, stunted and severely stunted individuals identified using the MEIRU wallchart and the correlation with the gold standard assessment. Using the MEIRU wallchart, six individuals (3.8%) of normal height-for-age were misclassified as stunted; these individuals had HAZ that were close to the HAZ<-2 cut-off for stunting, ranging from -2.00 to -1.83. The MEIRU wallchart misclassified two stunted individuals as non-stunted (3.8%) and similarly, the HAZ of these individuals were close to the HAZ<-2 cut-off (-2.08 and -2.04).

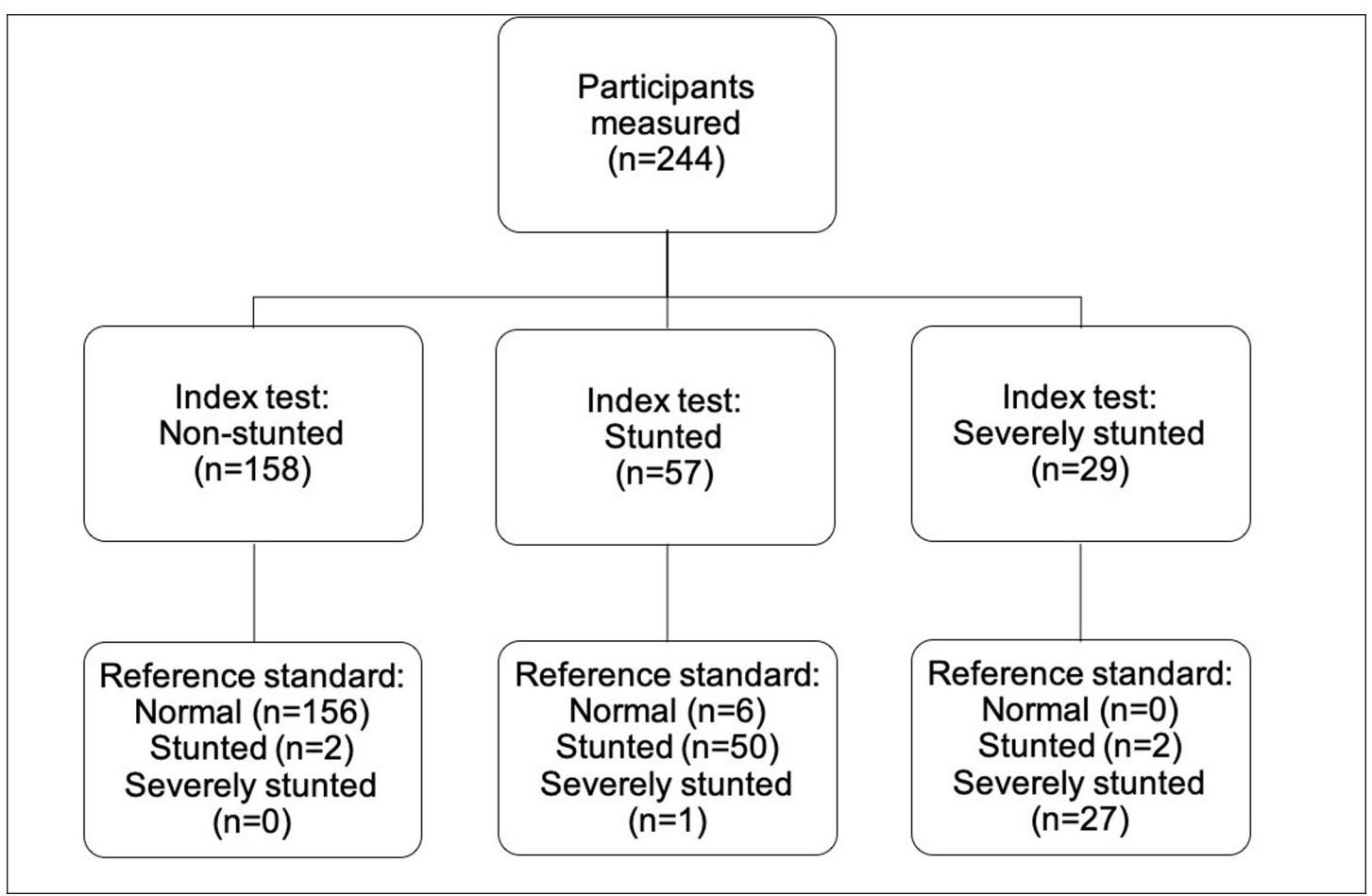

**Fig 6. Participant flow diagram.**

In comparison, use of the WHO lookup table misidentified 62 non-stunted individuals (38.3%) as stunted or severely stunted, and had one missed diagnosis of a stunted individual (1.9%) (Table B in S2 Text). Using the WHO growth charts (Table C in S2 Text) resulted in misclassification of 40 (28.8%) non-stunted individuals as stunted and missed 23 individuals with stunting (30.3%), seven of whom were severely stunted (25.9%).

**Estimates of diagnostic accuracy.** Fig 7 illustrates overall and category-specific agreements of each method with gold standard HAZ. The MEIRU wallchart had an overall agreement of 95.5%. WHO lookup tables and growth charts had overall agreements of 59.4% and 61.9% respectively, but there was large variability in the agreements within each category. Both traditional methods had very low "stunted" percent agreement of 31.5% for lookup tables and

**Table 1. Stunting classification of adolescents, by test method.**

| Stunting status | Gold standard calculated HAZ (n = 244) | MEIRU wallchart (n = 244) | WHO lookup table (n = 244) | WHO growth chart (n = 215) |
|---|---|---|---|---|
| | n (%) | n (%) | n (%) | n (%) |
| Normal (HAZ ≥ -2) | **162 (66.4)** | 158 (64.8) | 101 (41.4) | 122 (56.7) |
| Stunted (HAZ < -2) | **54 (22.1)** | 57 (23.4) | 63 (25.8) | 47 (21.9) |
| Severely stunted (HAZ < -3) | **28 (11.5)** | 29 (11.9) | 80 (32.8) | 46 (21.4) |

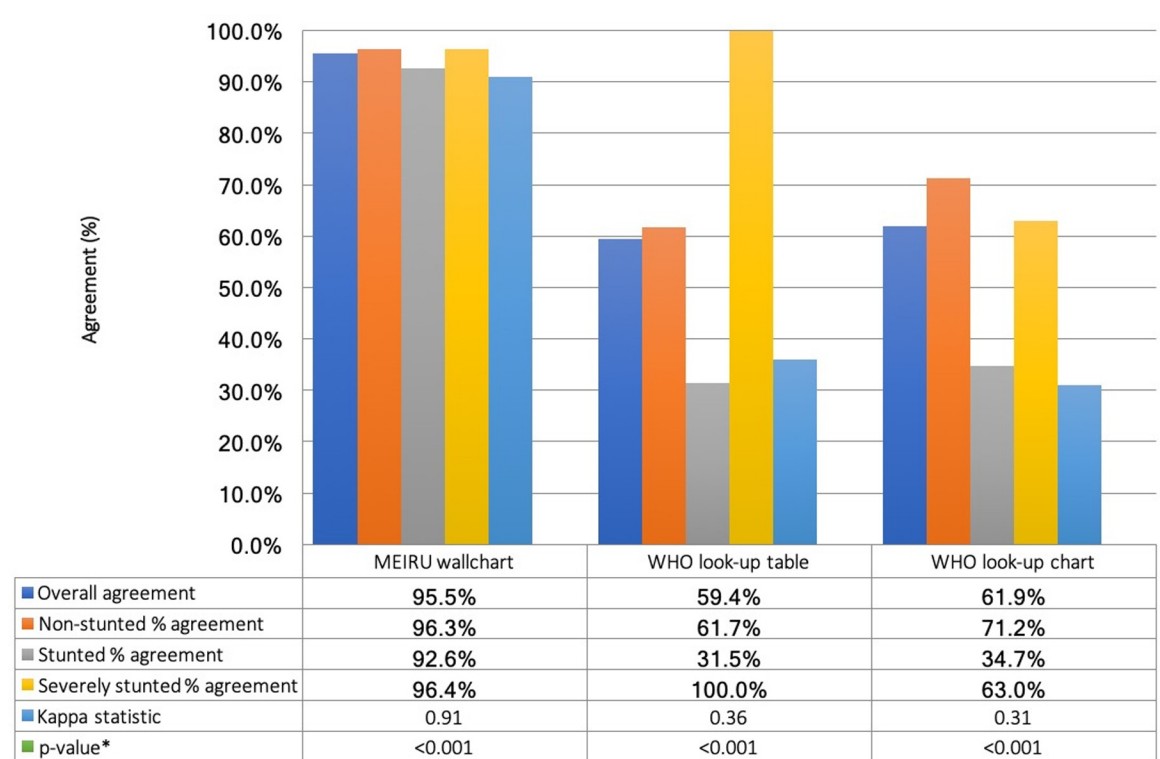

**Fig 7. Percentage agreement and kappa statistic of each test method, compared to gold standard HAZ.** *z-test of kappa statistic.

34.7% for growth charts. Although all severely stunted participants were correctly identified using the WHO lookup tables, 9.9% (16/162) of non-stunted individuals and 66.7% (36/54) of stunted individuals were misdiagnosed as severely stunted (Table B in S2 Text).

Table 2 show markers of diagnostic accuracy of each method at the HAZ<-2 and HAZ<-3 cut-offs. These include sensitivity, specificity, PPV, NPV, and agreement. At the HAZ<-2 (stunting) cut-off, the MEIRU wallchart had a sensitivity of 97.6% (95%CI: 91.5,99.7), specificity of 96.3% (95%CI: 92.1,98.6), and agreement of 96.7% (kappa = 0.93). The sensitivity, specificity, and agreement of the MEIRU wallchart at the HAZ<-3 cut-off (severe stunting) were 96.4% (95%CI: 81.7,99.9), 99.1% (95%CI: 96.4,99.9), and 98.8% (kappa = 0.94) respectively.

**Table 2. Diagnostic accuracy of each test method around HAZ-2 (stunting) and HAZ <-3 (severe stunting) cut-offs.**

| Test method | Sensitivity, % (95% CI) | p-value* | Specificity, % (95% CI) | p-value* | PPV, % (95% CI) | NPV, % (95% CI) | Agreement, % | Kappa |
|---|---|---|---|---|---|---|---|---|
| **Stunting (HAZ <-2)** | | | | | | | | |
| MEIRU wallchart (n = 244) | 97.6 (91.5, 99.7) | - | 96.3 (92.1, 98.6) | - | 93.0 (85.4, 97.4) | 98.7 (95.5, 99.8) | 96.7 | 0.93 |
| WHO lookup table (n = 244) | 98.8 (93.4, 100) | 0.564 | 61.7 (53.8, 69.2) | <0.001 | 56.6 (48.1, 64.9) | 99.0 (94.6, 100) | 74.2 | 0.51 |
| WHO growth chart (n = 215) | 69.7 (58.1, 79.8) | <0.001 | 71.2 (62.9, 78.6) | <0.001 | 57.0 (46.3, 67.2) | 81.1 (73.1, 87.7) | 70.7 | 0.39 |
| **Severe stunting (HAZ <-3)** | | | | | | | | |
| MEIRU wallchart (n = 244) | 96.4 (81.7, 99.9) | | 99.1 (96.7, 99.9) | | 93.1 (77.2, 99.2) | 99.5 (97.4, 100) | 98.8 | 0.94 |
| WHO lookup table (n = 244) | 100 (87.7, 100) | 0.317 | 75.9 (69.7, 81.5) | <0.001 | 35.0 (24.7, 46.5) | 100 (97.8, 100) | 78.7 | 0.42 |
| WHO growth chart (n = 215) | 63.0 (42.4, 80.6) | 0.003 | 84.6 (78.6, 89.4) | <0.001 | 37.0 (23.2, 52.5) | 94.1 (89.4, 97.1) | 81.9 | 0.37 |

*McNemar's $\chi^2$ test comparing sensitivity and specificity of each method against the MEIRU wallchart

At both HAZ-2 and -3 cut-offs, the McNemar's $\chi^2$ test comparing the WHO lookup tables against the MEIRU wallchart showed strong evidence of discordance between the specificities (p<0.001), but not for sensitivities (p = 0.564 at HAZ-2; p = 0.317 at HAZ-3). When the WHO growth charts were compared against the MEIRU wallchart, there was strong evidence of discordance between the sensitivities and specificities at both cut-offs (p<0.001).

Diagnostic performance markers based on individual HSAs for all methods are presented in Table A-C in S3 Text. The MEIRU wallchart performed very well for most HSAs. In contrast, the WHO lookup tables and growth charts did not perform as well, and this was the case for over half of the HSAs. The number of adolescents measured by each HSA was not sufficiently large for useful confidence interval calculations, or for more robust analyses of inter-HSA reliability and differences.

**Time taken for measurement.** Overall assessment using the MEIRU wallchart took an average of 24.8 (SD 10.6) seconds. Using lookup tables if was 66.2 (SD 28.0) seconds. There is difference of 41.4s or 62.5% (p<0.001). Of the four assessment steps, height measurement and the use of lookup tables account for most of the excess time (Fig 8).

**Acceptability and preference.** Overall, both participants and healthcare workers strongly preferred the MEIRU chart over the traditional methods. 71.5% of all respondents (83.3% of HSAs; 70.8% of adolescents) preferred the MEIRU wallchart to the traditional methods. 91.7% of HSAs and 90.0% adolescents found the MEIRU wallchart fast to use. 91.7% of HSAs and 88.1% of adolescents liked the method. Fig A, B in S4 Text show details of reported preferences as reported in a Likert scale.

HSAs were asked for more detailed feedback on the MEIRU chart. Overall, it was well-received and the preferred method over traditional HAZ assessment methods (Fig 9).

The main reasons for preference of the MEIRU wallchart were the speed of measurement (49.4%) and aesthetics of the wallchart (15.8%). Those who preferred the traditional method liked it for the speed of measurement (36.5%) and because it provided a height measurement in addition to a stunting diagnosis (30.2%).

Also related to acceptability of stunting assessment we note that both HSAs and adolescents saw stunting as an important issue. 66.7% of HSAs and 52.6% of adolescents saw it as an important or very important issue. Only 16.7% and 34.9% saw it as unimportant (Fig A in S5 Text).

Adolescents were overall less worried than HSAs about being identified as stunted– 54.5% were not worried or not worried at all while 33.3% were worried or very worried. However, only 21.5% of adolescents and 16.7% of HSAs said a person would be very worried by being identified (Fig B in S5 Text). Finally, 41.7% of HSAs and 47.9% of adolescents would feel embarrassed or very embarrassed if they were identified as stunted (Fig C in S5 Text).

## Discussion

Our novel, appropriate-technology MEIRU wallchart performed well in identifying stunted adolescents, with a very high overall agreement of 95.5% (kappa = 0.91, p<0.001) against gold standard HAZ. This indicates almost perfect agreement, as classified by Landis and Koch [18]. It also performed strongly in other measures of diagnostic performance–sensitivity, specificity, PPV and NPV. In contrast, performance of the two traditional measures, look-up chart and growth chart was poor, especially with regards to specificity. As well as performing better as a 'test' of stunting, the MEIRU chart was quicker than the two traditional methods, saving a statically significant mean of 41.4 seconds per assessment. In the context of a busy clinic or survey where many individuals need to be assessed this is likely also clinically significant. Finally, the MEIRU wallchart was strongly preferred by both participants and healthcare workers. Assessing stunting was acceptable to the majority of both healthcare workers and children/

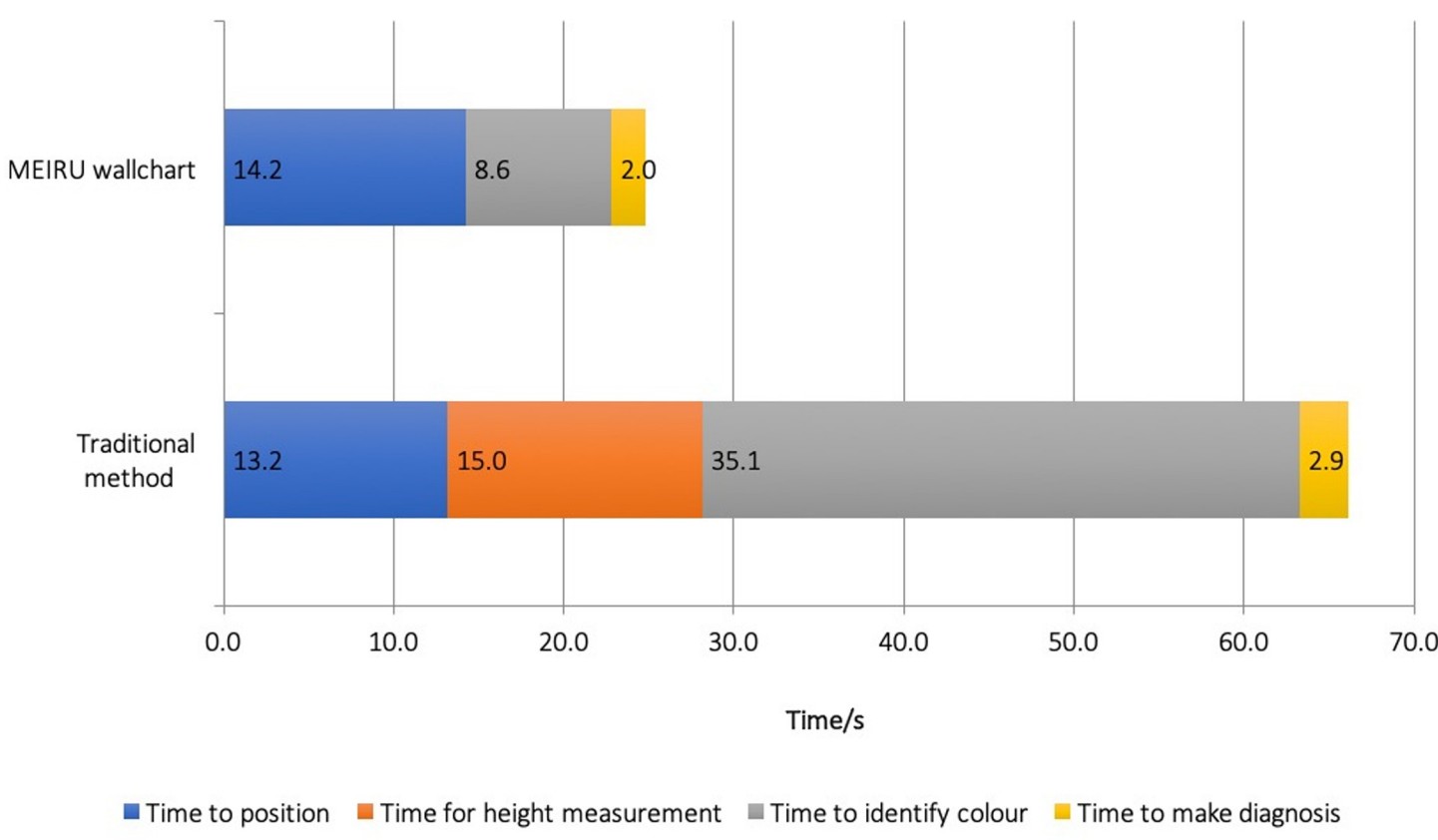

**Fig 8. Breakdown of time taken for each measurement step.** (n = 243).

adolescents. In combination, these results suggest that the MEIRU wallchart is a valuable screening tool with marked advantages over traditional, currently used alternatives [21].

Assessment of stunting is not simple. Some sources of assessment error are common to all the methods used: age unknown or incorrectly calculated; shoes not removed; hairstyle falsely adds height; incorrect measurement technique or poor posture resulting in wrong height measurement. However, traditional assessment methods involve more steps and since each step has potential for error, the overall likelihood of error/incorrect stunting category is markedly greater: error in reading exact height measure; error when consulting look-up table (e.g. wrong row read); error when plotting height on growth chart; error translating the HAZ score into a category of stunting; The frequency of errors using traditional methods was surprising, especially given the fact that HSAs are already familiar with these methods. The WHO lookup tables and growth charts, which are currently used in clinical practice, had overall category agreements with calculated HAZ of only 59.4% and 61.9% respectively. This was largely due to errors in interpreting lookup tables and growth chart. These poor performances were due to mistakes in using the lookup tables and growth charts to ascertain final stunting status. Given the number of errors observed during this study, it is likely that errors are also common in clinical and survey settings.

## Strengths and limitations

We are the first to both develop and test the MEIRU wallchart. Rather than focus on electronic assessment aids, which are popular but have their own limitations and risks [22], we took an

## HSA questionnaire responses

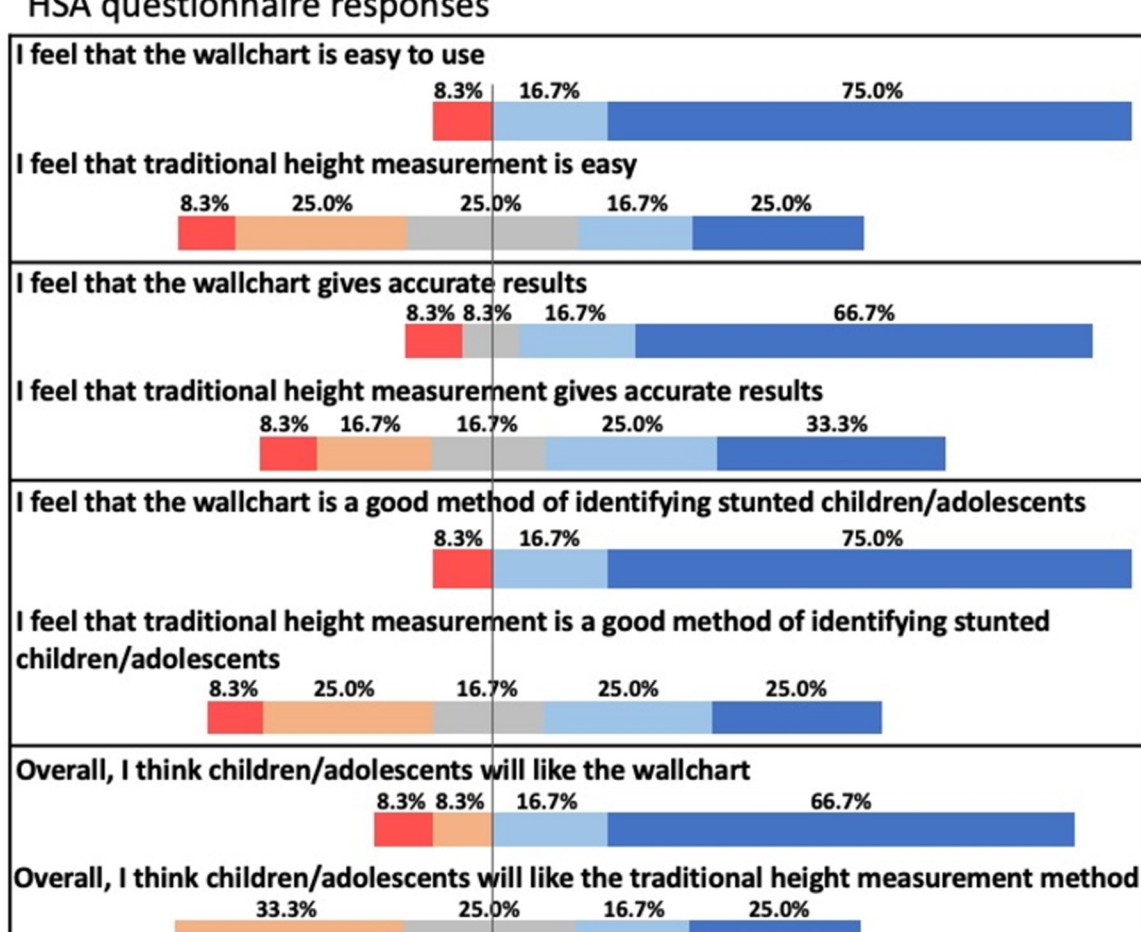

**Fig 9. HSAs' perspectives on acceptability and ease of use of the MEIRU wallchart (n = 12).**

'appropriate technology' approach whereby the MEIRU wallchart is "affordable, decentralized, energy-efficient, environmentally sustainable, and locally autonomous" [16, 23]. It can easily be printed and reproduced. Our results show that it gives a quick and easy classification of non-stunted; stunted; severely stunted. A strength of our study design is our study setting. Results are likely to be generalisable in similar settings where HSAs or other health and non-health workers doing the assessment have similar educational backgrounds, training and prior expertise. This warrants further confirmatory testing since we cannot exclude setting-specific effects. Future testing should also include more healthcare workers so that the effects of prior training and years of experience can be formally examined and quantified.

We acknowledge a number of limitations. First, we focused only on the final interpretation of stunting status. We did not explore the effect of additional errors that would be introduced by incorrect age ascertainment and incorrect height measures. Traditional assessments usually involve just a single measure of height, to the nearest mm, using a stadiometer. For a valid,

precise, gold standard measure, we minimised these errors by using the WHO two-observer method [13]. In routine clinical practice where this approach is not used, errors and inaccuracies would thus be even more common than we observed [24].

Second, it was not possible to blind our study: whilst surprising that HSAs performed so poorly using traditional methods they were familiar with, it is possible that they paid greater attention to the MEIRU wallchart due to its novelty. This is a factor that may wear off in more routine use.

Third, we acknowledge that the MEIRU wallchart also has potential for different error which were avoided in this study due to our correct initial setup. For instance, not positioning the chart properly on the floor could lead to either under- or over-reading of height. We used a portable version printed on canvas and propped up by wooden poles–but future versions might just be painted on a clinic wall to minimise these errors–assuming the initial painting is correctly done.

Fourth, the MEIRU chart performed much better than our initial sample size assumptions and an updated sample size calculation at the interim analysis, required 954 adolescents to maintain 80% power (S1 Text). We did not have the resources or capacity for this. However, our study is not underpowered and we do not believe that our results represent a false positive finding in favour of the MEIRU chart. A different method for calculating sample size for studies evaluating new diagnostic methods based on calculations around sensitivity and specificity gave sample sizes of $n^* = 83$ for sensitivity of 97.6% and $n^* = 100$ for specificity of 96.3% [25].

Finally, our prototype MEIRU wallchart as tested here focused on children and adolescents aged 8 and above. This is because the project was nested in a wider study focusing on adolescent stunting. In contrast, much clinical and public health work on stunting focuses on children aged under 5 years and in particular on infants aged under 2 years. There is no reason why an expanded version of the chart would not work just as well with additional bars extending down to younger age groups. This would though need to be formally tested. In particular, a different version of the chart would be needed for very young children aged <2 years since these are traditionally measured lying down rather than standing.

## Implications for practice, recommendations for future research

It is useful to consider possible individual and population uses of the MEIRU wallchart separately:

**Individual.** Due to the complex and time-consuming nature of using lookup tables and growth charts for linear growth assessment, individual assessment of height-for-age or stunting is rarely done in busy and resource-poor settings. Whilst we have compared the MEIRU chart against these traditional methods, a truer conceptualization of the problem might be using the MEIRU wallchart vs. doing no assessment at all. Stunted children and adolescents are widely missed and their potential needs are rarely met. Some would argue that this is not a problem since stunting is a complex, multifactorial issue which is often misinterpreted and misused as a measure of child health [26]. There is certainly no argument for routine screening to identify *nutritional* stunting as per Wilson-Junger criteria. Key criteria for such a programme are *not* currently met:

- **there should be an accepted treatment recognised for the stunting condition**: this is not currently the case for nutritional stunting [27]. Given the multifactorial nature of the problem (associated factors range from suboptimal nutrition to underlying illness to pathogen-contaminated environments) there is unlikely to even be one simple, single solution to the problem of global stunting.

- **there should be a policy on who should be treated:** most current global policy/programme focus is on the 'first 1000' days as a particularly sensitive period of growth and development [6, 28]. Though it could easily be adapted for these younger infants, the current version of the MEIRU chart focuses on older children and adolescents. Some data but limited data suggests that these older ages represent an important 'second window' of opportunity to tackle stunting and associated problems [9, 29].

- **diagnosis and treatment should be cost-effective:** this evidence is lacking

- **treatment should be more effective if started early / there should be a recognisable latent or early symptomatic stage** it is biologically plausible that offering any future treatment(s) to mildly stunted children before they become severely stunted is a good thing and results in better outcomes. But no evidence on this issue is currently available.

Future work might change the above. If new effective and cost-effective new interventions do arise, it is important to recognise the Wilson-Jungner conditions that *are* met even now:

- **the condition should be an important health problem:** though a complex condition [30], stunting is without doubt a major global health problem [5, 31, 32].

- **the natural history of the condition should be understood**: despite many evidence gaps, long term health and development adversities associated with stunting are well described [31].

- **there should be an acceptable test that is easy to perform and interpret, acceptable, accurate, reliable, sensitive and specific:** our data suggest that the MEIRU chart can play this role much better than look-up tables or growth charts.

Even though routine clinical screening for stunting is not currently indicated, there may be other roles for the MEIRU chart in clinical settings. There are for example a number of medical conditions which manifest with short stature or stunting. Best known is growth hormone deficiency [33, 34] but there are many others including renal disease and coeliac disease [35]. Short stature may be the only immediate manifestations. Future research needs to quantify how common such treatable conditions are in LMIC settings. The MEIRU chart may have a role in an initial clinical assessment to help identify them.

**Population.** We fully agree with authors who highlight the limitations of stunting as a measure of child health and call for it to be interpreted with care [26]. This does not however mean it should not be measured or used at all. Use at population level is particularly valuable. Despite being an imperfect measure of health and nutrition, it can be and is a useful measure of health and nutritional status–including in older children and adolescents [8]. As such, stunting prevalence is important to know. It can inform and guide local, national and international policy and practice. Current surveys are time consuming and costly to carry out and analyse. In future, the MEIRU wallchart might help with rapid surveys. It gives an immediate result without need for calculations and can be used within the community by minimally-trained community members. A version can easily be painted on or pinned to the walls in places like health centres and hospitals for assessing populations there. It might even be used in non-health settings by schools, teachers or older students to conduct measurements. With a better understanding of the scale of stunting in a local area, communities might initiate new ideas for prevention and improvement of general health and nutrition. The MEIRU wallchart might thus be used as a teaching and health promotion aid to raise awareness about stunting in the community.

Future research is needed to explore such population based and survey uses. Research in a variety of different settings with a variety of different users would be important. Part of this

research would be to further explore any social stigma around stunting. Our results did not suggest immediate cause for concern about stigma but this could be very different in other settings. Education programmes, if necessary, will be essential in targeting these concerns.

## Conclusions

The MEIRU wallchart is an acceptable and appropriate screening tool for detection of stunted and severely stunted adolescents in the community. Its' screening test performance is excellent with high sensitivity and specificity. Whilst there is no current justification for routine screening for nutritional stunting, the MEIRU wallchart may have a role in future programmes; in surveys; in specific clinical assessments. In contrast, two common current assessment methods (lookup tables and growth charts) performed poorly and should therefore be used with caution: extra training and supervision is needed with these.

## Supporting information

**S1 Checklist. STARD checklist.**
(DOCX)

**S1 Table. Table of baseline characteristics of adolescents in the study.**
(DOCX)

**S1 Text. Sample size calculation based on different possible misclassification rates.**
(DOCX)

**S2 Text. Tables showing cross-tabulation of stunting status from HAZ and stunting status using different methods (MEIRU wallchart, WHO lookup tables, WHO growth charts).**
(DOCX)

**S3 Text. Diagnostic accuracies of each test method, by HAS.**
(DOCX)

**S4 Text. Questionnaire responses on speed of each method and individual preferences.**
(DOCX)

**S5 Text. Questionnaire responses on perception of stunting in the community and individual views on stunting.**
(DOCX)

**S6 Text. PLOS' questionnaire on inclusivity in global research.**
(PDF)

## Acknowledgments

We thank the children, adolescents and healthcare workers who took part in this work and made it possible. We also thank the wider MEIRU team hosting and supporting this project.

## Author Contributions

**Conceptualization:** Pannapat Amy Chanyarungrojn, Natasha Lelijveld, Amelia Crampin, Lawrence Nkhwazi, Steffen Geis, Moffat Nyirenda, Marko Kerac.

**Data curation:** Pannapat Amy Chanyarungrojn.

**Formal analysis:** Pannapat Amy Chanyarungrojn.

**Funding acquisition:** Pannapat Amy Chanyarungrojn, Marko Kerac.

**Investigation:** Pannapat Amy Chanyarungrojn, Natasha Lelijveld, Amelia Crampin, Lawrence Nkhwazi, Steffen Geis, Moffat Nyirenda, Marko Kerac.

**Methodology:** Pannapat Amy Chanyarungrojn, Natasha Lelijveld, Amelia Crampin, Lawrence Nkhwazi, Steffen Geis, Moffat Nyirenda, Marko Kerac.

**Project administration:** Amelia Crampin, Lawrence Nkhwazi, Steffen Geis.

**Resources:** Amelia Crampin, Steffen Geis, Marko Kerac.

**Supervision:** Natasha Lelijveld, Amelia Crampin, Moffat Nyirenda, Marko Kerac.

**Validation:** Pannapat Amy Chanyarungrojn, Natasha Lelijveld, Amelia Crampin, Lawrence Nkhwazi, Steffen Geis, Moffat Nyirenda, Marko Kerac.

**Visualization:** Pannapat Amy Chanyarungrojn.

**Writing – original draft:** Pannapat Amy Chanyarungrojn.

**Writing – review & editing:** Pannapat Amy Chanyarungrojn, Natasha Lelijveld, Amelia Crampin, Lawrence Nkhwazi, Steffen Geis, Moffat Nyirenda, Marko Kerac.

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
