## [Decision Letter · Decision Letter 0]

11 Apr 2023

PGPH-D-23-00105

Tools for assessing child and adolescent stunting: Lookup tables, growth charts and a novel appropriate-technology “MEIRU” wallchart - a diagnostic accuracy study

Dear Dr. Chanyaungrojn,

Thank you for submitting your manuscript to PLOS Global Public Health. After careful consideration, we feel that it has merit but does not fully meet PLOS Global Public Health’s publication criteria as it currently stands. Therefore, we invite you to submit a revised version of the manuscript that addresses the points raised during the review process.

We look forward to receiving your revised manuscript.

Kind regards,

Dickson Abanimi Amugsi, PhD

Academic Editor

Journal Requirements:

1. Please include a complete copy of PLOS’ questionnaire on inclusivity in global research in your revised manuscript. Our policy for research in this area aims to improve transparency in the reporting of research performed outside of researchers’ own country or community. The policy applies to researchers who have travelled to a different country to conduct research, research with Indigenous populations or their lands, and research on cultural artefacts. The questionnaire can also be requested at the journal’s discretion for any other submissions, even if these conditions are not met.  Please find more information on the policy and a link to download a blank copy of the questionnaire here: https://journals.plos.org/plosone/s/best-practices-in-research-reporting. Please upload a completed version of your questionnaire as Supporting Information when you resubmit your manuscript.

a) Please clarify all sources of financial support for your study. List the grants, grant numbers, and organizations that funded your study, including funding received from your institution. Please note that suppliers of material support, including research materials, should be recognized in the Acknowledgements section rather than in the Financial Disclosure. 

b) State the initials, alongside each funding source, of each author to receive each grant. For example: "This work was supported by the National Institutes of Health (####### to AM; ###### to CJ) and the National Science Foundation (###### to AM)."

c) State what role the funders took in the study. If the funders had no role in your study, please state: “The funders had no role in study design, data collection and analysis, decision to publish, or preparation of the manuscript.”

d) If any authors received a salary from any of your funders, please state which authors and which funders.

3. Please update your online Competing Interests statement. If you have no competing interests to declare, please state: “The authors have declared that no competing interests exist.”

4. Some material included in your submission may be copyrighted. According to PLOS’s copyright policy, authors who use figures or other material (e.g., graphics, clipart, maps) from another author or copyright holder must demonstrate or obtain permission to publish this material under the Creative Commons Attribution 4.0 International (CC BY 4.0) License used by PLOS journals. Please closely review the details of PLOS’s copyright requirements here: PLOS Licenses and Copyright. If you need to request permissions from a copyright holder, you may use PLOS's Copyright Content Permission form.

Potential Copyright Issues:

Figure 1: Please confirm whether you drew the images / clip-art within the figure panels by hand. If you did not draw the images, please provide (a) a link to the source of the images or icons and their license / terms of use; or (b) written permission from the copyright holder to publish the images or icons under our CC-BY 4.0 license. Alternatively, you may replace the images with open source alternatives. See these open source resources you may use to replace images / clip-art:

- " ext-link-type="uri" xlink:type="simple">https://openclipart.org/"

Figures 4 and 5 contain branding/a logo. We are not permitted to publish this under our CC-BY 4.0 license, even with permission. We ask that you please remove or replace it.

Additional Editor Comments (if provided):

Thank you for submitting your work to PGPH for publication. It has received favourable reviews from two independent reviewers. I suggest you carefully address their comments and resubmit for consideration.

I have some minor issues I would like you to address:

1. Multiple in-text references should be in one parenthesis, e.g. [15, 25, 45]. In case references are in sequence, e.g. 24,25,26,27,28,29 should be written as [24-29].

2. Include standard deviation (SD) in the binary form of the z-scores. e.g. HAZ-2SD etc. This correction should be made throughout the manuscript.

Good luck in the revision.

Reviewers' comments:

Reviewer's Responses to Questions

**Comments to the Author**

1. Does this manuscript meet PLOS Global Public Health’s publication criteria? Is the manuscript technically sound, and do the data support the conclusions? The manuscript must describe methodologically and ethically rigorous research with conclusions that are appropriately drawn based on the data presented.

Reviewer #1: Yes

Reviewer #2: Yes

2. Has the statistical analysis been performed appropriately and rigorously?

Reviewer #1: Yes

Reviewer #2: Yes

3. Have the authors made all data underlying the findings in their manuscript fully available (please refer to the Data Availability Statement at the start of the manuscript PDF file)?

Reviewer #1: No

Reviewer #2: Yes

4. Is the manuscript presented in an intelligible fashion and written in standard English?

Reviewer #1: Yes

Reviewer #2: Yes

5. Review Comments to the Author

Reviewer #1: This is a good attempt at introducing innovation to the measurement of child health outcomes. The advantages of the proposed methods over the traditional ones should be relevant for policy especially in dealing with the lags issues between the publication of demography surveys. However, the external validity of the proposed method could be improved upon if consistent results are obtained across: developing countries; and other child health outcomes (wasting, underweight). Are findings sensitive to context? Also, it is unclear on which theoretical framework is the methodology for measurement based? can the study be more clear on how guidance are followed in the development of the measurement index?

Reviewer #2: First I congratulate the authors for coming up with this innovative work of method to assess the stunting o children and adolescent groups. The manuscript is well written, data is well presented however, there few comments which need to be addressed as per attachment.

Reviewer Comments:

LINE DESCRIPTION

66 Remove repeated word :… We recruited 244 recruited.

235 It is good to used simple available materials.., however I doubt if the use of canvas held by two wooden sticks is stable enough for accurate readings. Will you also suggest printing on a soft board for stability and for accurate readings?

240 and 242 Do they stand in the same manner as in stadiometer/length board?, i.e., do the same rules apply?

318 You mentioned only data for males, can you also mention the data girls as well?.

326 244 is your total sample, if so, I think you should instead of n=244 it should read (N=244)

422 Insert space after 41.1

425 Recheck for punctuation (full stop) between adolescents and In combination…

411 and 442 Indicate what solutions to be taken as it is highly needed to minimize these errors in order to give MEIRU more value.

462 May be you need to Rephrase the sentence by adding the word approach. “In routine clinical practice where this approach is not used, “

470 and 471 Rephrase the sentence to read: “For instance, by not positioning the chart properly on the floor could lead to either under- or over-reading of heights”.

472 I agree to have it painted on the wall or on a soft board for stability among other factors! In future what about a child 2yrs who cannot stand upright? Considering in surveys and clinics all children need to be attended also for their stunting status. Link with sentence in line 488-490

504 i) Contextualizing (translating into a local situation) proved frameworks like UNICEF conceptual framework for malnutrition can ease the burden.

ii) there should be an accepted treatment recognized for the disease: Note: since nutritional stunting is not a disease rather it is a condition; then I suggest you rephrase the sentence. ..”there should an accepted approach/strategy to correct the stunting condition”

Otherwise this work is a good innovation! With modifications/improvements it can be scaled-up to replace the traditional methods especially at community level.

6. PLOS authors have the option to publish the peer review history of their article (what does this mean?). If published, this will include your full peer review and any attached files.

**Do you want your identity to be public for this peer review?** For information about this choice, including consent withdrawal, please see our Privacy Policy.

Reviewer #1: No

Reviewer #2: No

---

## [Decision Letter · Decision Letter 1]

14 Jun 2023

Tools for assessing child and adolescent stunting: Lookup tables, growth charts and a novel appropriate-technology “MEIRU” wallchart - a diagnostic accuracy study

PGPH-D-23-00105R1

Dear Dr Chanyarungrojn, 

We are pleased to inform you that your manuscript 'Tools for assessing child and adolescent stunting: Lookup tables, growth charts and a novel appropriate-technology “MEIRU” wallchart - a diagnostic accuracy study' has been provisionally accepted for publication in PLOS Global Public Health.

Best regards,

Dickson Abanimi Amugsi, PhD

Academic Editor

Reviewer Comments (if any, and for reference):

Reviewer's Responses to Questions

**Comments to the Author**

1. If the authors have adequately addressed your comments raised in a previous round of review and you feel that this manuscript is now acceptable for publication, you may indicate that here to bypass the “Comments to the Author” section, enter your conflict of interest statement in the “Confidential to Editor” section, and submit your "Accept" recommendation.

Reviewer #2: All comments have been addressed

2. Does this manuscript meet PLOS Global Public Health’s publication criteria? Is the manuscript technically sound, and do the data support the conclusions? The manuscript must describe methodologically and ethically rigorous research with conclusions that are appropriately drawn based on the data presented.

Reviewer #2: Yes

3. Has the statistical analysis been performed appropriately and rigorously?

Reviewer #2: Yes

4. Have the authors made all data underlying the findings in their manuscript fully available (please refer to the Data Availability Statement at the start of the manuscript PDF file)?

Reviewer #2: Yes

5. Is the manuscript presented in an intelligible fashion and written in standard English?

Reviewer #2: Yes

6. Review Comments to the Author

Reviewer #2: No further comments

7. PLOS authors have the option to publish the peer review history of their article (what does this mean?). If published, this will include your full peer review and any attached files.

**Do you want your identity to be public for this peer review?** For information about this choice, including consent withdrawal, please see our Privacy Policy.

Reviewer #2: No
